# Effect of Solid-State Fermentation Products of *Lactobacillus plantarum*, *Candida utilis*, and *Bacillus coagulans* on Growth Performance of Broilers and Prevention of Avian Colibacillosis

**DOI:** 10.3390/vetsci11100468

**Published:** 2024-10-01

**Authors:** Fangfang Li, Bing Lv, Jiakun Zuo, Saqib Nawaz, Zhihao Wang, Liyan Lian, Huifang Yin, Shuming Chen, Xiangan Han, Haidong Wang

**Affiliations:** 1College of Veterinary Medicine, Shanxi Agricultural University, Jinzhong 030801, China; kybgs408@163.com (F.L.); lvbing0117@163.com (B.L.); 2Shanghai Veterinary Research Institute, The Chinese Academy of Agricultural Sciences (CAAS), 518 Ziyue Road, Shanghai 200241, China; jkzuo1214@163.com (J.Z.); nawazsaqib143@gmail.com (S.N.); wangzh2048@163.com (Z.W.); 18016565087@163.com (L.L.); 3Engineering Research Center for the Prevention and Control of Animal Original Zoonosis, Fujian Province, College of Life Science, Longyan University, Longyan 364012, China; yin197608@163.com

**Keywords:** *Lactobacillus plantarum*, *Candida utilis*, *Bacillus coagulans*, avian pathogenic *Escherichia coli*

## Abstract

**Simple Summary:**

Probiotics have a range of effects on poultry industry, including improving production performance, inhibiting or mitigating pathogenic infections, enhancing antioxidant capacity, reducing toxic effects, and improving the intestinal environment of poultry. They have significant potential for use as an ideal alternative to antibiotics in poultry farming. The purpose of this research was to evaluate the effect of solid fermentation products of *Lactobacillus plantarum*, *Candida utilis*, and *Bacillus coagulans* on the growth characteristics of broilers and the prevention of avian colibacillosis. The findings indicate that solid fermentation products can enhance the growth characteristics, immune function, and intestinal morphology of broilers. Additionally, it can optimize the abundance of microflora in the cecum and prevent avian colibacillosis, which is caused by avian pathogenic *Escherichia coli* (APEC).

**Abstract:**

This study investigates the impact of the solid-state fermentation products of *Lactobacillus plantarum*, *Candida utilis*, and *Bacillus coagulans* (LCBs) on the growth characteristics, immune function, intestinal morphology, cecum microbial community, and prevention of avian colibacillosis in broilers. One hundred and twenty Hyland Brown broilers (aged one day) were divided randomly into three groups (four replicates of ten broilers per group). (1) The CON group was fed a basal diet. (2) The MOD group was fed a basal diet. On day 40, APEC strain SX02 (1.1 × 10^5^ CFU/g) was administered to the breasts of chickens in this group. (3) The LCBs group was fed a basal diet supplemented with fermentation products (98.5% basal diet + 0.5% *Lactobacillus plantarum* and *Candida utilis* solid-state fermentation products + 1.0% *Bacillus coagulans* solid-state fermentation products). On day 40, the LCBs group received the same treatment as the MOD group. The experiment lasted 43 days. This study found that the average daily gain (ADG) of the LCBs group was significantly higher than that of the MOD group (*p* < 0.05), indicating that LCBs can significantly increase the ADG of broilers and improve the feed conversion ratio. Furthermore, compared to the MOD group, the heart bacterial load was significantly reduced in the LCBs group (*p* < 0.05), and the lesions less severe in the heart, liver, and jejunum were observed (*p* < 0.05). Additionally, the detection of intestinal flora showed a significant increase in the abundance of beneficial bacteria in the cecum of the LCBs group, while the number of *Escherichia coli* and *Shigella* decreased significantly. In conclusion, the solid fermentation of *Lactobacillus plantarum*, *Candida utilis*, and *Bacillus coagulans* can improve the growth performance of broilers while also protecting against avian pathogenic *Escherichia coli* infection. This demonstrates the potential usefulness of these LCBs in feed production.

## 1. Introduction

The excessive employment or unjustified utilization of antibiotics in animal feed inevitably results in a plethora of negative outcomes such as drug residues, the escalation of multidrug-resistant strains, environmental contamination, and disturbances to the intestinal flora and food safety, all of which significantly jeopardize the well-being of both humans and animals [1,2,3]. Avian colibacillosis, caused by avian pathogenic *Escherichia coli* (APEC), is an important bacterial disease that affects the poultry industry which leads to pericarditis, perihepatitis, swollen head syndrome, septic anemia, enteritis, synovitis, and cellulitis in all types of poultry production. As one of the principal causes of the morbidity and mortality in chickens, avian colibacillosis is associated with significant economic losses to the poultry industry [3,4]. APEC is difficult to prevent and control due to its complex serotypes and extensive drug resistance, resulting in a lack of clinically effective drugs and vaccines, posing a serious threat to the poultry industry [5]. Probiotics are microorganisms that can promote host health when appropriately administered, often displaying probiotic properties such as bacterial inhibition, immune system enhancement, and improved gut flora [6,7]. Solid fermentation describes the process by which microorganisms ferment on a solid medium with minimal or no free water. This fermentation technique is known for its low cost, short cycle, simple implementation, and environmental sustainability [8,9]. Therefore, probiotic solid fermentation has the potential to be an alternative to antibiotics and an affordable feed additive.

*Lactobacillus plantarum* produce organic acids and bacteriocins, which are antimicrobial metabolites that can lower the pH of the animal’s gut and inhibit the proliferation of feed and gut-associated pathogenic bacteria [10]. *Lactobacillus plantarum MB001* microencapsulation could potentially be used as a feed additive to improve cecal microbiota, gut integrity, and nutrient utilization, leading to improved performance in broilers [11]. Embryo-injected *L. plantarum PA01* altered the structure and metabolites of the microbial flora before and after hatching, specifically promoting Lactobacillus colonization [12]. *Candida utilisis* produce a variety of nutrients, including proteins, B vitamins, minerals, and enzymes. This strain can utilize both five- and six-carbon sugars for growth, as well as inexpensive by-products from industry and agriculture to synthesize proteins. Therefore, *Candida utilis* is an excellent candidate for producing proteins for food, medicine, or animal feed. Studies have shown that the metabolites produced by yeast promote the growth of lactic acid bacteria, suggesting that the two are symbiotic, and the mechanism may be that the yeast provides the lactic acid bacteria with the nutrients needed for growth [13,14]. *Candida utilis* can use nitric acid as a nitrogen source, grow without the addition of growth factors in the medium, and produce human-edible proteins using molasses. Therefore, *Candida utilis* is a high-quality yeast product [15]. *Bacillus coagulans* could improve the production performance of laying hens, improve the intestinal flora, and increase the height of the ileal villus [16]. When *Bacillus coagulans* was used with *Lactobacillus plantarum*, the relative abundance of carbohydrate metabolism, energy metabolism, cofactors, and vitamin metabolism was increased, and the relative abundance of amino acid metabolism and drug resistance was decreased [17]. *Bacillus coagulans* is considered an edible and safe strain approved as a feed additive [18]. *Bacillus coagulans XY2* can modulate biosorption, intestinal microflora bacteria, and lipid metabolism [19]. More importantly, *Bacillus coagulans* is particularly acid-tolerant; it adapts to the acidic environment of the stomach and reaches the intestine where it exerts its probiotic effect [20,21].

Our previous studies characterized the properties of *Lactobacillus plantarum, Candida utilis,* and *Bacillus coagulans* (LCBs) and optimized the fermentation process, resulting in the optimal formulation of solid fermentation products of *Lactobacillus plantarum*, *Candida utilis*, and *Bacillus coagulans*. In this study, we investigated the changes in LCBs on chick growth performance; organ pathological lesions; organ indices; heart, liver, and lung bacterial load; and cecal microflora. The aim was to further investigate the effect of LCBs in improving chick growth performance and preventing avian pathogenic *Escherichia coli* (APEC), which results in serious losses to the poultry industry around the world, and to provide a theoretical basis for probiotic preparations as feed additives.

## 2. Materials and Methods

### 2.1. Bacterial Strains, Broilers, and Diets

*Lactobacillus plantarum*, *Candida utilis*, *Bacillus coagulans*, and avian pathogenic *Escherichia coli* (APEC SX02) were provided by the Laboratory of Fermentation Engineering, Shanxi Agricultural University. The LCBs were produced by the solid-state fermentation of *Lactobacillus plantarum*, *Candida utilis*, and *Bacillus coagulans*. The detailed composition of LCBs is listed in Table 1. In the fermentation products used in this study, the viable numbers of *Bacillus coagulans, Lactobacillus plantarum,* and *Candida utilis* were 2.1 × 10^8^, 1.1 × 10^9^, and 2.0 × 10^8^ CFU/g, respectively.

The complete fermentation process and ingredients are described in Appendix A.

Liquid fermentation parameters for *Bacillus coagulans*: 0.75 g of red jujube corn flour, 0.75 g of peanut powder, and 0.025 g of K2HPO_4_ were added into 50 mL of ultrapure water, and then mixed well. After autoclaving and sterilization, 2.5 mL of *Bacillus coagulans* seed liquid was added (the viable number of *Bacillus coagulans* was 1.0 × 10^8^ CFU/mL) into it, mixed well, and incubated at 37 °C 160 r/min for 24 h. Under these conditions, the viable count of *Bacillus coagulans* reaches 9.1 × 10^8^ CFU/mL. The parameters for solid-state fermentation of *Bacillus coagulans* using sterilized wheat bran are as follows: 80 mL of ultrapure water and 8 mL liquid fermentation medium were added to 100 g of distilled dry bran, mixed well, and incubated at 37 °C for 24 h. The viable numbers of *Bacillus coagulans* were 2.1 × 10^8^ CFU/g.

Liquid fermentation parameters for *Lactobacillus plantarum* and *Candida utilis*: 3.16 g of red jujube corn flour, 2.02 g of peanut meal, and 0.19 g of Na2HPO_4_ were added into 50 mL ultrapure water, mixed well. After autoclaving and sterilization, 2.5 mL of Lactobacillus plantarum and Candida utilis seed liquid (1:1) was added into it (the viable numbers of *Lactobacillus plantarum* and *Candida utilis* were 1.0 × 10^9^ CFU/mL and 1.0 × 10^8^ CFU/mL, respectively), mixed well, and incubated at 37 °C 160 r/min for 24 h. The parameters for solid-state fermentation of *Lactobacillus plantarum* and *Candida utilis* using sterilized wheat bran are as follows: 80 mL of ultrapure water and 8 mL of liquid fermentation medium were added to 100 g of sterile dry bran, mixed well, and incubated at 37 °C for 24 h. The viable numbers of *Lactobacillus plantarum* and *Candida utilis* were 1.1 × 10^9^ CFU/g and 2.0 × 10^8^ CFU/g.

The LCBs group was fed a basal diet supplemented with fermentation products (98.5% basal diet + 0.5% *Lactobacillus plantarum* and *Candida utilis* solid-state fermentation products + 1.0% *Bacillus coagulans* solid-state fermentation products). The composition and nutritional composition of the basal diet are detailed in Appendix A. One hundred and twenty Hyland Brown broilers (aged one day) were randomly assigned to three groups (four replicates of ten broilers per group). The experiment lasted 43 days. On day 40, *APEC SX02* (1.1 × 10^5^ CFU/g) was administered to the breasts of chickens in the MOD and LCBs groups, and the CON group was replaced with saline [22]. The animal experiment schematic diagram is described in Figure 1.

### 2.2. Sample Collection

When the broilers were 43 days old, tissue samples were collected from 20 chickens per group. Blood was collected from the heart and centrifuged at 4 °C and 4000 r/min for 10 min, and the serum was separated and stored in a −20 °C refrigerator for backup. The heart, liver, spleen, and bursa of Fabricius were collected. The heart, liver, and lung lesions were taken from the relevant parts, fixed in 4% paraformaldehyde, and stored at −80 °C. Then, 1 g of the heart, liver, and lung was aseptically collected in a centrifuge tube. About 2 cm of jejunal tissue was taken and fixed in 4% paraformaldehyde. Then, 1 g of the duodenum, jejunum, and cecum contents was aseptically collected into centrifuge tubes and stored at −80 °C.

### 2.3. Determination of Heart, Liver, Spleen, and Bursa of Fabricius Organ Indices

The dissected heart, liver, spleen, and bursa of Fabricius were weighed and the organ index was calculated with the following formula: organ index = organ weight (g)/body weight (kg) [23].

### 2.4. Determination of Macroscopic Lesions and Histomorphology of the Heart, Liver, and Jejunum

Each group randomly selected 20 samples. A macroscopic observation of the heart and liver mainly includes the appearance, size, shape, color, texture and accompanying symptoms of organs. In addition, For the jejunal villus length and crypt depth is detected. Details of the scoring system to (semi)quantify the severity of lesions about the liver and heart as follows.

0-grade (1 point): The organ appears normal with no signs of pathological changes.

1-grade (2 points): Mild pathological changes, such as slight congestion, edema, or scattered petechial hemorrhages, with no apparent impact on organ function.

2-grade (3 points): Moderate pathological changes characterized by more extensive involvement, including moderate congestion, edema, hemorrhage, or focal necrosis, which may have some effect on organ function.

3-grade (4 points): Severe pathological changes with widespread and pronounced involvement, including massive hemorrhage, diffuse necrosis, and structural damage, significantly affecting organ function.

4-grade (5 points): Extremely severe pathological changes where the organ has nearly lost all functionality, manifesting as extensive hemorrhage, necrosis accompanied by infection, and potentially leading to the death of the mouse.

### 2.5. Collection of Tissue Samples for Histology Assessment

Visceral tissues in 4% paraformaldehyde were fixed overnight at 25 °C. After dehydration by ethanol gradient, xylene transparency, and soaking in paraffin overnight, the tissue was cut into 5 μm thick sections. Samples were deparaffinized and placed in xylene I; xylene II; 50% (*v*/*v*) xylene/ethanol; and 100, 90, 75, and 50% ethanol for 3 min before hematoxylin and eosin (H&E) staining. Finally, the samples were kept in pure water until subsequent staining. H&E staining was performed using an H&E reagent (Servicebio, Wuhan, China) according to the manufacturer’s instructions [24]. The specimens were read blinded. First, the pathological sections to be evaluated were uniformly numbered to ensure that the numbering was independent of the source of the specimen. Secondly, all slices were treated in the same way (such as fixation, dehydration, slicing, staining, etc.) to reduce the impact of treatment differences on the evaluation results. Finally, three pathologists observed and evaluated each slice without knowing the source of each slice. The evaluation included cell morphology, tissue structure, lesion degree, and immunohistochemical markers.

### 2.6. Determination of Heart, Liver, and Lung Bacterial Load

The heart, liver, and lung tissues were separated into centrifuge tubes under aseptic conditions. The tissues were weighed, ground, and diluted with a gradient of tissue homogenate. The samples were diluted 10 times. Then, three dilutions were chosen. Take 0.2 mL of each into a flat dish. Pour in the appropriate amount of culture medium melted and cooled to about 45 °C, mix with the bacterial solution, cooling, to become solidified, and place in the incubator at 37 °C. After 24 h, count the colonies. Calculate the number of bacteria in the original solution using this formula: average number of colonies × dilution × 5 [25].

### 2.7. Molecular Analysis of Microbial Communities

The samples were randomly selected, and each group was tested 6 samples. Genomic DNA was extracted using a FastDNA^®^ SPIN Kit (Tiangen, DP302-02, Tiangen, China). The DNA quality was verified by an ultramicro spectrophotometer (Thermo, NanoDrop 2000, Waltham, MA, USA). The DNA used for high-throughput sequencing was amplified using primers 341F-CCTACGGGNGGCWGCAG and 805R-GACTACHVGGGTATCTAATCC, targeting the V3-V4 regions of 16S rRNA. The PCR products were detected by agarose gel electrophoresis. The target products were purified and recovered using the Gum Recovery Purification Kit and quantified and sequenced using the Qubit 3.0 DNA detection kit with the Illumina HiSeq platform. The 16SrRNA high-throughput sequencing analysis was performed on broiler cecum microorganisms.

### 2.8. Statistical Analysis

A One-way ANOVA was performed using SPSS 22.0, Duncan’s multiple comparisons test, and the independent samples *t*-test. The Box–Behnken design was used with Design-Expert 12.0. Graphing was performed with GraphPad Prism 7.0. All data are expressed as the mean ± SD of at least three independent experiments. Duncan’s multiple comparisons test was performed to analyze the average daily feed intake, the average daily gain, organ index, and nutrient composition. The least significant difference (LSD) test was used for analyses of bacterial load. The ANOVA was used for the analyses of the composition of the microbiota. *p* < 0.05 was considered significant.

### 2.9. Use of AI or AI-Assisted Technologies

DeepL Write AI Technology was used to improve the English tenses, grammar, and stylistic aspects of the writing in this article.

## 3. Results

### 3.1. Growth Performance

The average daily feed intake (ADFI) of the LCBs group was significantly higher compared to the CON and MOD groups from one to forty-three days of age (*p* < 0.01) (Figure 2a). Compared with the CON and MOD groups, the average daily gain (ADG) of the LCBs group increased significantly (*p* < 0.01) (Figure 2b). In addition, in the CON, MOD, and LCBs groups, the effects of an APEC infection on the average daily feed intake (ADFI) and the average daily gain (ADG) of broilers from 40 to 43 days were determined (Appendix A), indicating that broilers with an APEC infection can significantly increase the ADFI and reduce the ADG. The survival rate of broilers in the LCBs group was significantly higher in the MOD group (Figure 2c). This suggests that LCBs have a positive effect on chick growth performance and against an APEC infection.

### 3.2. Organ Index

The liver, spleen, and heart indices were significantly higher in the MOD group than in the CON group, and the bursa index was significantly lower (*p* < 0.05), indicating that APEC infection caused damage to the heart, liver, spleen, and bursa of broilers. Compared with the MOD group, the liver index of broilers in the LCBs group was significantly lower *(p* < 0.05) and the bursa index was significantly higher (*p* < 0.05). The addition of LCBs to the diet reduced the damage to the liver and bursa caused by *APEC SX02* (Table 2).

### 3.3. Macroscopic Lesions of the Heart, Liver, and Jejunum

The pathological observations revealed that there were significant macroscopic lesions in the heart of the MOD group compared to the CON group, specifically in the form of thickening of the pericardium. Compared to the MOD group, the thickened pericardium was reduced in the LCBs group (Figure 3a–c). Compared with the CON group, the liver in the MOD group showed significant lesions, as evidenced by an enlargement, brittle texture, blunt rounded edges, pale color, and outer layer covered with fibrin film. Compared with the MOD group, the liver in the LCBs group showed significant remission, as evidenced by the reduction in size, significant improvement in color and texture, and the absence of fibrin film covering the outer layer (Figure 3d–f). Compared with the CON group, the jejunal segment of the MOD group was partially edematous. Compared with the MOD group, the jejunum edema in the LCBs group was reduced (Figure 3g–i).

### 3.4. Histomorphology of the Heart, Liver, and Jejunum

Compared with the CON group, myocardial fiber breakage, myocardial fiber spacing widening, myocardial outer membrane thickening, and inter-myocardial inflammatory cell infiltration were aggravated in the MOD group. Compared with the MOD group, myocardial fiber spacing widening, myocardial outer membrane thickening, and myocardial inter-inflammatory cell infiltration were significantly alleviated in the LCBs group (Figure 4a–c). In the MOD group, hepatic platelets were unevenly arranged, inflammatory cell infiltration increased, hepatocytes underwent significant degeneration and necrosis, and interstitial edema with congestion and hemorrhage occurred. In the LCBs group, the uneven platelet arrangement was slightly improved, and the inflammatory cell infiltration, hepatocyte degeneration and necrosis, interstitial edema congestion, and hemorrhage were all reduced (Figure 4d–f). The MOD group showed a shorter jejunal villi length and broken villi tips compared to the CON group. The LCBs group showed longer jejunal villi and alleviated tip breakage compared to the MOD group (Figure 4g–i).

### 3.5. Bacterial Load of Organisms

Compared with the MOD group, the LCBs group broilers had a significantly lower heart bacterial load (*p* < 0.05) and no significant changes in liver and lung bacterial load (*p* > 0.05), indicating that LCBs significantly reduced the APEC numbers in the hearts of diseased chickens (Figure 5).

### 3.6. Jejunal Villus Length and Crypt Depth

Jejunal villus length and crypt depth were significantly decreased in the MOD group compared with the CON group (*p* < 0.05), and villus length/crypt depth was not significant (*p* > 0.05). Compared with the MOD group, the LCBs group had a significantly elevated villi length and villi length/crypt depth, and a significantly decreased crypt depth (*p* < 0.05), indicating that LCBs had a significant effect on chicks. The LCBs group showed a significant increase in villus length and villus length/crypt depth and a significant decrease in crypt depth (*p* < 0.05) (Table 3).

### 3.7. Cecal Microbiota Composition

The Simpson and Shannon indices were not significant in the three groups (*p* > 0.05) (Figure 6a,b). Firmicutes, Bacteroidetes, Proteobacteria, and Desulfobacterota were the dominant flora (phylum level) in the CON group cecal. Compared with the CON group, the Bacteroidetes, Proteobacteria, and Desulfobacterota abundance in the MOD group was significantly decreased. Compared with the MOD group, the Firmicutes abundance was significantly decreased in the LCBs group. The Bacteroidetes and Proteobacteria abundance was significantly increased in the LCBs group (Figure 6c). Ruminococcaceae and Clostridia were the dominant flora (family level) in the CON group cecal. The Ruminococcaceae and Oscillospiraceae abundance of the LCBs group was lower than that of the MOD group, and the Rikenellaceae abundance of the LCBs group was higher than that of the MOD group (Figure 6d).

## 4. Discussion

APEC is capable of reducing the performance of broilers, thus causing significant economic losses to the farming industry. Hung et al. have shown that Bacillus coagulans can improve the growth performance of broilers by increasing feed conversion [26]. Tarabees R. et al. used a mixture of *Lactobacillus plantarum*, *Clostridium butyricum*, and an acidifier mixed with the base diet of broilers and showed that the mixture could prevent chicken *E. coli* disease by improving production performance and regulating the intestinal microflora [27]. In this study, the content of nutrients in the feed fed to chickens in the CON, MOD, and LCBs groups was detected, and there was no difference among the three groups (Appendix A). So the influence of differences in intake of nutrients on the growth and development of chickens was excluded. The MOD group was found to significantly reduce ADG and significantly increase ADFI and lethality in broilers, which is consistent with the findings of Zhang et al. [28]. This also proves that avian colibacillosis model is successful. Compared with MOD and CON groups, the solid-state fermentation products of *Lactobacillus plantarum*, *Candida utilis*, and *Bacillus coagulans* significantly increased ADG and improved the survival rate in broilers, suggesting that LCBs had the effect of enhancing chick performance and effectively preventing *E. coli*.

Based on microscopic lesions, the MOD group showed typical lesions of pericarditis and perihepatitis, indicating successful replication of the chicken *E. coli* disease model. Compared with the MOD group, the heart, liver, and jejunum lesions were significantly alleviated in the LCBs group, suggesting that LCBs had a protective effect on the heart, liver, and jejunum of broilers infected with APEC. From the H&E sections, it was also found that LCBs protected the heart by improving myocardial fiber breaks and inflammatory cell infiltration and reducing the thickness of the outer wall of the myocardium. In addition, it protected the liver by improving the uneven arrangement of liver platelets, inflammatory cell infiltration, interstitial edema, and congestion and bleeding of the liver; and it protected the jejunum by improving the jejunal villus tip breaks.

The typical pathological manifestations of APEC are pericarditis and perihepatitis [29]. The spleen and bursa of Fabricius are important immune organs in chickens [30]. The liver, spleen, and heart indices of the LCBs group were significantly higher than those of the other groups, and there was no significant difference between the organ indices of the MOD group. LCBs significantly reduced the liver index and heart load. They increased the bursa index, indicating that this solid fermentation product had a protective effect on the liver, bursa injury, and heart caused by APEC. This is similar to the findings of Wang et al. [31], where LCBs significantly reduced the bacterial load in the liver and colon of broilers suffering from chicken *E. coli* disease.

The length of intestinal villus and crypt depth are important indicators that reflect the digestive capacity of broilers. Wu et al. found that *Lactobacillus plantarum JM113* supplementation altered intestinal morphology, especially in the duodenum and jejunum, where villi were shorter and crypts were deeper in broilers [18]. *Bacillus coagulans* significantly increased the villous height (VH), crypt depth (CD), and VH:CD ratio in the jejunum and ileum of chickens [32]. In this study, LCBs were found to significantly increase the villi length and villi length/crypt depth, suggesting that LCBs can promote digestion and absorption in animals, which may be the reason why the ADG of the LCBs group was higher than that of the MOD group. The ADFI was lower than that of the MOD group.

Wang et al. found that a mixture of four strains of probiotics, including *Bacillus coagulans*, improved the intestinal flora by increasing the number of beneficial bacteria and decreasing the number of harmful bacteria [17]. Broiler intestinal microorganisms fluctuate the most from shelling to about 1 week after shelling, and the composition and diversity of microorganisms are unstable, which is the best time for manual intervention [33]. The results of this study showed that the dominant species of intestinal flora in the cecum of control broilers included the phylum Firmicutes, Bacteroidetes, and Proteobacteria, which is similar to the results of Xiao et al. [34]. They are known to maintain the health of the organism, increase the content of short-chain fatty acids in the cecum, and alleviate inflammation [35]. Although LCBs do not improve the diversity of the cecum intestinal flora, they can increase the abundance of the genera Alternaria and ruminal cocci, and in addition, LCBs reduce the abundance of cecum *E. coli-Shigella* spp. This may be because probiotics can compete with commensals and enteric pathogens for adhesion sites in the mucus layer or intestinal epithelial cells (IECs), thereby preventing harmful colonization and enhancing barrier function [36]. It is also possible that LCBs increase the abundance of the Bacteroidetes and other intestinal anaerobes in the cecum, thus reducing *E. coli* colonization in vivo.

The gut microbiota is a complex ecosystem, and there is competition between different types of bacteria. When feeding feed containing phylum Firmicutes, other types of bacteria or microorganisms may be introduced at the same time, and these newly introduced microorganisms may compete with the original phylum Firmicutes, thereby inhibiting their growth and reproduction. Some bacteria may produce antibiotic-like substances or other inhibitors that directly inhibit the growth of phylum Firmicutes, leading to a decline in their numbers in the gut. The research shows that probiotics and prebiotics can quickly reduce the phylum Firmicutes/Bacteroidetes ratio, inhibit harmful bacteria (such as *Klebsiella* and *Escherichia coli*), and accelerate the recovery of beneficial intestinal microorganisms (such as *Lactobacillus*) [37,38].

## 5. Conclusions

Conclusively, incorporating LCBs into the baseline diet for broilers effectively enhances the average daily weight gain of broilers; minimizes the liver index of ailing broilers; elevates the bursal index; diminishes the cardiac bacterial load; lessens histopathological damage to the heart, liver, and jejunum; increases the jejunal villus length; augments the ratio of villus length to crypt depth; and maintains the intestinal microflora. These findings illuminate the advantages of supplementing LCBs as an alternative to antibiotics in the feed of poultry.

## Figures and Tables

**Figure 1 vetsci-11-00468-f001:**
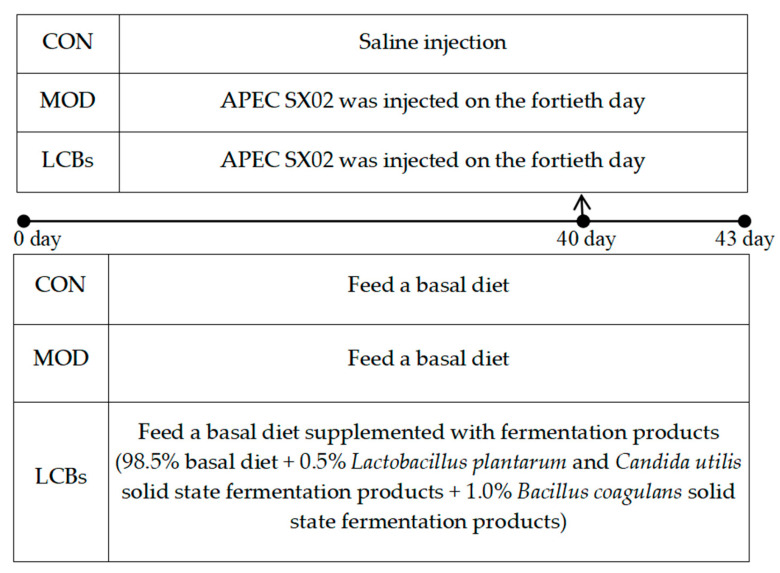
The animal experiment schematic diagram.

**Figure 2 vetsci-11-00468-f002:**
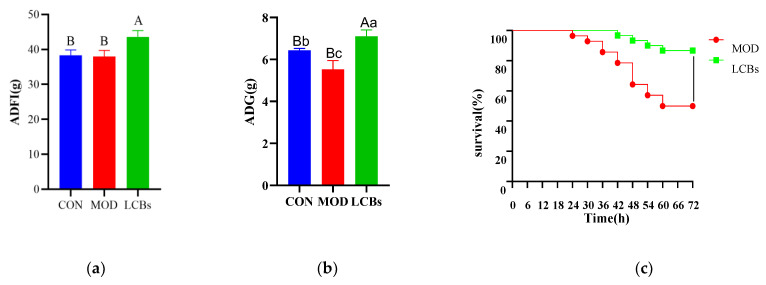
Growth performance. (**a**) The average daily feed intake of broilers from one to forty-three days of age. (**b**) The average daily gain of broilers from one to forty-three days of age. Data were analyzed using SPSS 22.0 ANOVA and the differences were compared using SPSS 22.0 Duncan’s multiple range test to the significance levels of 5% and 1%. Diverse lowercase letters show significant differences (*p* < 0.05), diverse capital letters show significant differences (*p* < 0.01). (**c**) The survival rate of broilers after APEC SX02 infection.

**Figure 3 vetsci-11-00468-f003:**
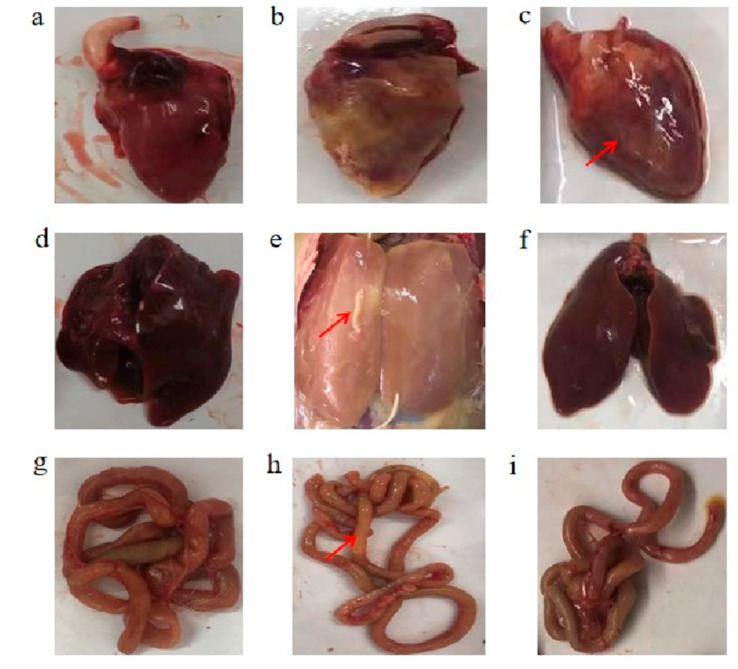
Microscopic lesions of the (**a**) CON heart, (**b**) MOD heart, (**c**) LCBs heart, (**d**) CON liver, (**e**) MOD liver, (**f**) LCBs liver, (**g**) CON jejunum, (**h**) MOD jejunum, and (**i**) LCBs jejunum were observed 3 days after *APEC SX02* infection. The red arrows indicate macroscopic lesions of the heart, liver, and jejunum, with congestion, edema and hemorrhage.

**Figure 4 vetsci-11-00468-f004:**
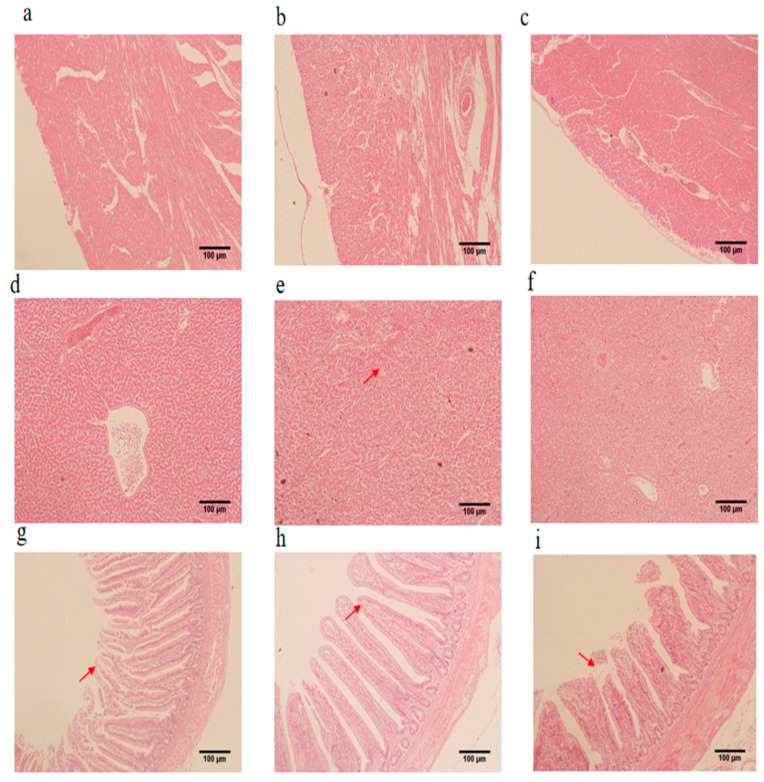
Tissue lesions of the heart, liver, and jejunum. Tissue morphology of (**a**–**c**) CON heart, MOD heart, and LCBs heart, (**d**–**f**) CON liver, MOD liver, and LCBs liver, and (**g**–**i**) CON jejunum, MOD jejunum, and LCBs jejunum, respectively. The red arrows indicate a change in the tissue histomorphology of the heart, liver, and jejunum

**Figure 5 vetsci-11-00468-f005:**
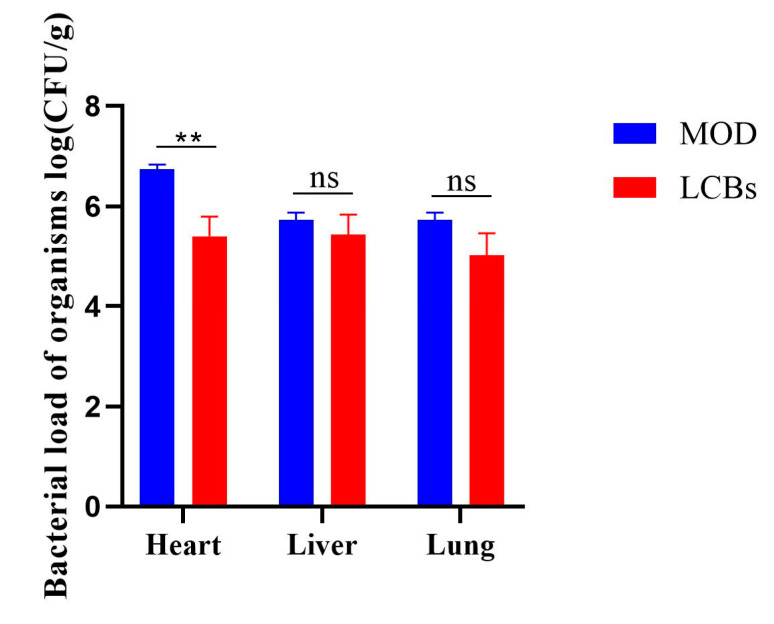
Bacterial load of heart, liver, and lung in CON, MOD, and LCBs groups. (Error bars are the mean ± SD, ** *p* < 0.01, ns means no significant difference, *n* = 3.)

**Figure 6 vetsci-11-00468-f006:**
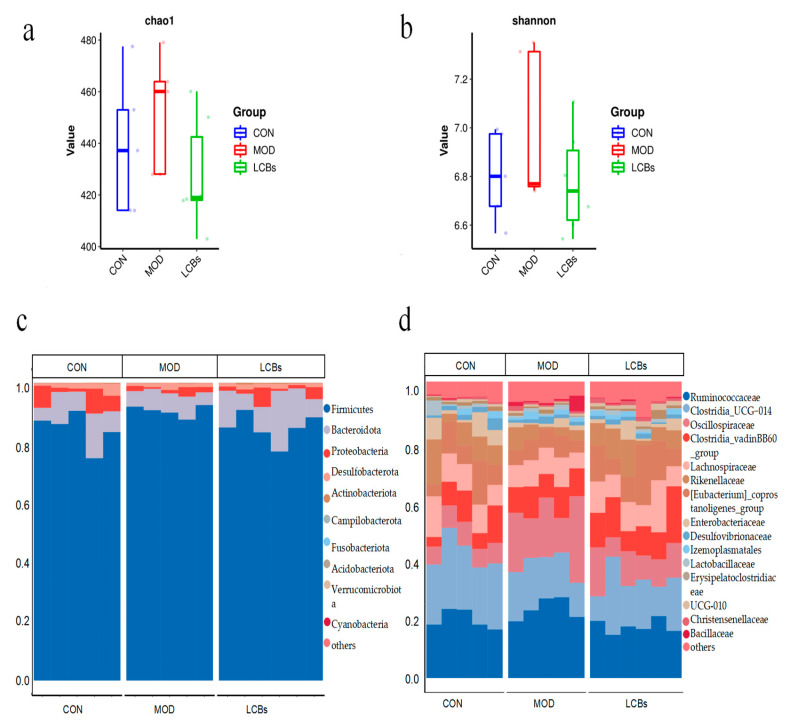
(**a**) Simpson index and (**b**) Shannon index of broilers after APEC SX02 infection. The composition of the intestinal flora at the (**c**) species and (**d**) family levels.

**Table 1 vetsci-11-00468-t001:** Composition of *Bacillus coagulans* preparation, and *Lactobacillus plantarum* and *Candida utilis* preparation.

Types	*Bacillus coagulans* Preparation	*Lactobacillus plantarum* and *Candida utilis* Preparation
Dry wheat bran (g)	100	100
Tap water (mL)	80	80
Liquid fermentation medium (mL)	8	8
Fermentation conditions	Under natural conditions (temperature was 25–30 °C, humidity was 35–40%), fermentation for 1–3 days	Under natural conditions (temperature was 25–30 °C, humidity was 35–40%), fermentation for 2–6 days
Spore, CFU/g	>10^8^	>10^9^, >10^7^

**Table 2 vetsci-11-00468-t002:** Determination of organ index.

Group	Liver Index	Bursal Index	Heart Index	Spleen Index
CON	28.02 ± 2.62 ^c^	1.43 ± 0.21 ^b^	8.75 ± 1.29 ^c^	3.37 ± 0.74 ^a^
MOD	55.39 ± 5.76 ^a^	3.89 ± 0.78 ^a^	14.83 ± 3.99 ^a^	1.83 ± 0.91 ^c^
LCBs	45.07 ± 5.23 ^b^	4.43 ± 0.77 ^a^	13.05 ± 3.62 ^ab^	2.19 ± 0.72 ^ab^

Note: Means with different lowercase superscripts in the same column differ significantly (*p* < 0.05).

**Table 3 vetsci-11-00468-t003:** Determination of jejunal villus length and crypt depth.

Group	Villi Height (μm)	Crypt Depth (μm)	Villi Height/Crypt Depth
CON	376.39 ± 16.28 ^a^	48.45 ± 3.79 ^ab^	7.81 ± 0.81 ^b^
MOD	332.30 ± 21.29 ^b^	44.77 ± 3.14 ^b^	7.43 ± 0.47 ^b^
LCBs	374.58 ± 21.22 ^a^	37.24 ± 2.94 ^c^	10.09 ± 0.71 ^a^

Note: Means with different lowercase superscripts in the same column differ significantly (*p* < 0.05).

## Data Availability

The datasets used during this study are available from the corresponding author on reasonable request.

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
