# Peer review of "Effect of Solid-State Fermentation Products of Lactobacillus plantarum, Candida utilis, and Bacillus coagulans on Growth Performance of Broilers and Prevention of Avian Colibacillosis"

_vetsci, 2024, doi:10.3390/vetsci11100468_

Round 1
Reviewer 1 Report (Previous Reviewer 2)
Comments and Suggestions for Authors
Probiotics have many positive effects on poultry growth performance and disease prevention. Through the rational use of probiotics, the efficiency and safety of poultry farming can be improved to meet the needs of consumers for high-quality and safe food. This study found that Effect of solid-state fermentation products of Lactobacillus plantarum, Candida utilis and Bacillus coagulans on growth performance of broilers and prevention of avian colibacillosis. It has important significance for the development of probiotic feed additives. However, there are still many small details that need to be modified in the article, in addition, if you want to make probiotic products, further in-depth research is needed.
1. Line 42: Delete “are” before“less severe”.
2. Line 47: Change “whilst also” for “while also”.
3. Line 62: Change “causes for” for “causes of”.
4. Line 92: Please add “ and ” before “ increase the height of the ileal villus ”.
5. Line 167: Should“twenty chickens per group”be amended to “20 chickens per group”?
6. Line 183: The descriptions of villus hight/crypt depth are inaccurate.
7. Line 211:Grammatical tenses should be unified. Such as“mixed with the bacterial solution, cooling……”Please check the grammatical tenses carefully again.
8. Line 212: change “placed” for “place”.
9. Line 216: Samples of 20 chickens were collected above, and only 6 samples were tested here, please explain.
10. Feed usually contains large numbers of other bacteria. Did you make any further microbiological investigation of the product after adding probiotic fermentation?
11. Line 239-243:You can hardly use the term improvement here. In what way has the conditions improved? Is there little effect on growth performance?
12. Line 267: What does “signicant” mean? If it is “significant”, please check the spelling carefully.
13. Line 342-344:The numbers of Firmicutes are lower in the LCBs group. But both Lactobacillus plantarum and Bacillus coagulans belong to the Firmicutes, Why?
14. Line 367: Please change “survival rate” to “the survival rate”.
15. When writing about Lactobacillus plantarum, Candida utility, Bacillus coagulans, and E. coli, please check italics, including references.
Author Response
Please see the attachment.

Reviewer 2 Report (New Reviewer)
Comments and Suggestions for Authors
The authors explored the effect of solid-state fermentation products of some probiotics on the growth of broilers. The study is well-designed and the results are well-presented and discussed. The solid-state fermentation products effectively decreased the avian colibacillosis caused by Escherichia coli, which shows its application prospects. Here are some tips to improve the manuscript:
1. Line 97-99: the authors mentioned their previous studies, please provide the reference.
2. Can you add a schematic illustration of the fermentation process to make it clearer to readers?
3. The animal experiment description should be more detailed. You can put the content of Figure S1 in the main text.
Author Response
Please see the attachment.

Reviewer 3 Report (New Reviewer)
Comments and Suggestions for Authors
The problem of excessive use of antibiotics has resulted in intensive search for novel strategies for microbial prevention in all areas of human activity, including food production. Different probiotics and fermentation products have gained lots of research interest due to their antibacterial activities and beneficial properties on immune system and intestinal microflora. The presented study is of a great interest because it explores the potential antibiotic effects of solid-state fermentation products of three microorganism on broilers focusing on their growth/weight gain and changes in intestinal morphology of heart, liver and intestinal organs. The potential of preventing avian colibacillosis, common bacterial disease severely affecting poultry industry, is also assessed, presenting the potential replacement for widely used antibiotics in chicken breeding industry.
The manuscript is very well written and organized. Results are clearly presented and discussed in regard to the already published research. I do not have any major objections or issue with the manuscript.
The authors present in table 1 the composition of the probiotic preparation but it would also be good to analyse the composition of fermentation products (e.g. HPLC) to identify potential active components.
Could the authors please comment on the statement “forbidden” in line 70 in the Introduction?
In the 2.5. title of the section, I suggest replacing the words “bacterial samples” since, in my understanding, the collection of tissue samples is described for latter analysis of bacterial presence.
Line 203, what does “…the target products were purified using gel.” means?
In the discussion, line 338, I suggest rephrasing the sentence in …the significant reduction of ADG and significant increase of… in MOD group was observed…, because, written in the way it is it suggests that the MOD group was the cause of the reduction and increase.
Author Response
Please see the attachment.

This manuscript is a resubmission of an earlier submission. The following is a list of the peer review reports and author responses from that submission.
Round 1
Reviewer 1 Report
Comments and Suggestions for Authors
This study investigates the effects of LCBs on chick growth characteristics and their potential to mitigate avian colibacillosis. The findings suggest that LCBs contribute to an increase in the average daily gain of chickens and offer protection against APEC infection. While the study holds significance in shedding light on the potential benefits of LCBs in poultry farming, several issues warrant attention
1. It would be beneficial to include a schematic diagram, in Figure 1, illustrating the experimental groups, treatment timing, and sample collection points, given the complexity of the animal experiments conducted in this study.
2. It is noteworthy that LCBs demonstrate a protective effect against mortality in chicks. However, it is observed that there are no significant alterations in the bacterial load within the liver and lung tissues. Considering the known inflammatory response induced by APEC, as evidenced by previous studies (PMID: 34399296), it is recommended to conduct qPCR experiments to compare the mRNA levels of proinflammatory cytokines TNF-α, IL-1β, and IL-6 in the heart, liver, and lung among the CON, MOD, and LCBs groups. This would elucidate how LCBs regulate inflammation in the APEC infection model.
3. It appears that Figure 1a and Figure 1b may depict contradictory information based on the description of your results.
4. Figure 3 lacks a legend explaining the meaning of labels such as a-i. Including a detailed legend would enhance the clarity of the figure.
5. The name of Figure 4 is the same as Figure3.
6. In Figures 5c and 5d, the text on both the x-axis and y-axis appear to be too small.
7. In lines 58-59, given the significance of your study, it is recommended to cite a more directly relevant study to demonstrate the negative impact of APEC on the poultry industry, such as PMID: 30157463.
Comments on the Quality of English Language1. Please ensure consistency in formatting for bacterial concentrations (e.g., 1.1 × 10^5 CFU/g) throughout the manuscript, which should be superscript, lines 34, 115, 116, 117, 134. And both italicized and non-italicized “APEC” appear in the text, and should be written uniformly in non-italicized.
2. Line 34, please add "." after "the MOD group".
3. In line 195, please remove the space before "APEC" to ensure proper formatting.
Reviewer 2 Report
Comments and Suggestions for Authors
Probiotics have gained considerable attention as an alternative to AGPs in poultry feeds. In this work, Li et al. reported the effect of solid-state fermentation products of Lactobacillus plantarum, Candida utilis and Bacillus coagulans on growth performance of chicks and prevention of avian colibacillosis. Some results are very meaningful. The manuscript is well written and can be read quite fluently. However, before publication, Some minor issues need to be corrected.
There are some minor issues which need to be corrected.
1. Check italics when write Lactobacillus plantarum, Candida utilis, Bacillus coagulans and Escherichia coli , such as line 120, 121.
2. Please use the spacer properly, like there needs a spacer between number and measuring unit.
3. Line 26: change “which caused by” for “which is caused by”.
4. Line 34: Please add “the” before “LCBs”.
5. Line 34: Should 105 be amended to 105?
6. Line 68: Therefore, probiotic solid fermentation has the potential to become an affordable "forbidden" feed additive. This sentence is ambiguous and does not conform to the meaning to be expressed, whether to change "forbidden" to "licensed" or others.
7. Line 88: EnFXs appears for the first time and abbreviations are not allowed.
8. Line 93: LCBs appears for the first time in the main text, should it be changed to Lactobacillus plantarum, Candida utilis and Bacillus coagulans (LCBs)?
9. Line 105: Please change “resulting” for “results”.
10. Line 110: Please change “escherichia coli” to “Escherichia coli”.
11. Line 114:Should 2.1 × 108 be amended to 2.1 × 108?
12. Line 115:Should 1.1 × 109 be amended to 1.1 × 109?
13. Line 116:Should 2.0 × 108 be amended to 2.0 × 108?
14. Line132: Please add “a” before “significant difference”.
15. Line 198: Please delete “,” before “The”.
16. Line 234: Please change “fibre” for “fiber”.
Reviewer 3 Report
Comments and Suggestions for Authors
L32, First and for most, the biggest concern of the reviewer is the very low replicates included in the experiment (only three replicates). This would largely harm the statistical power and increase variability.
L33, the authors mentioned four groups while only three were mentioned. What is the fourth group?
L57, What does the authors mean beasts here? This is a very inappropriate choice of word when the authors are conducting study in chickens which is a livestock. The authors should have English reviewed before submitting.
L78-83, What is the rationale for the introduction on Saccharomyces cerevisiae. This is not added in the tested products and not mentioned anywhere else again in the paper.
L86, change this to broilers studies, as the present study was in broilers.
L109-135, Where is the information about the diets? What is the inclusion rate of LCB in the diets. How was the diet formulated? For how long were the birds fed the experimental diets? How were the birds being fed before the APEC challenge?
L132 What is the broiler breed? And where were the birds obtained?
L133-135. What is the rationale for induction of colibacillosis this way. Is there previous studies reporting the success of induction of colibacillosis by this method?
L137, at 43 days, the birds can not be referred as chicks. How many birds were the tissues collected from please specify.
L139, what is bursa of Fasciola??? Is the authors referring to bursa of fabricius?
L147, when and why is fascial collected? I don't see the reason for collecting fascial in the present study.
L149-156, this section was describing the histology assessment but the title is transcriptional analysis.
L163-164, from where was the DNA extracted from?
L179-181, the reviewer appreciate the authors' honesty about the usage of AI tool. However, it does appear to the reviewer that the authors may have written the manuscript in their native language first and then use the AI to translate it into English which explained the very weird choice of words such as "beast" and "fascial", and "bursa of Fasciola".
L183, the study lasted 43 days, but only the last 72 hour performance was reported? Where is the results for the fist 40 days.
L183, how was the performance data statistically analyzed? The authors mentioned Duncan comparison then why use the T-test to compare between groups again? This increase the Type 1 error. Please report the Duncan test results not the T-test results.
L183, why was the most important indices FCR not reported? Also the final body weight should be presented here.
L221 was there a quantify score for assessment of these lesions? How many birds were this scoring performed on?
L265 Figure title is tissue lesion but the figure is showing bacteria load.
Figure 5 c and d. I don't see any difference in the results presented in these two figures. They are both microbiota at the level of phyla. It is phyla not family. Where is the species results?
L336-338, yes from hatch to 1 week the microbes changes the most but the study investigate the microbiome on day 43. How is this discussion related to the results?
Comments on the Quality of English LanguageThe English writing has very serious problem including but not limited to the poor choices of words. The inclusion of extensively long sentences.
These problems disturbingly suggested an overuse of the AI tool. Although the authors mentioned in the manuscript AI tool was used to improve the English writing, I doubt if the authors even checked the content generated by the AI as some of the mistakes are very blatantly obvious.
Round 2
Reviewer 1 Report
Comments and Suggestions for Authors
The authors have addressed my concerns. The revised version has shown significant improvement. I think this paper is now suitable for accepted for publication.
Author Response
Thank you for your guidance on my manuscript. I wish you good health and success in your work!
Reviewer 3 Report
Comments and Suggestions for Authors
The authors have carefully revised the manuscript based on the reviewer's comments. Some minor changes are needed.
L190, the treatment allocation details need to be added here. How many birds in total, birds per treatment and replicates. (Same as in the abstract). Also definition of MOD, LCB, and CON needs to be specify here for their first appearance.
L205 and L206. It is not fascia, it is bursa of Fabricius.
L274, remove extremely
L273-277, the way the authors presented the growth performance data was very confusing. The reviewer could not understand why the only ADFI of day 40-43 was assessed and reported. However, ADG was assessed from day 0-43? Corresponding results need to be provided (ADFI and ADG both from day 40-43; and ADFI and ADG both from day 0-43), otherwise the current way of presenting makes no sense and did not provide enough information. For the future reference, if the authors want to looks at treatment effects on growth performance, feed conversion ratio (FCR) is considered one of the most important index and should be included.
In Tables and Figures, P-values need to be listed. In a separate row for the tables and within the figure for the figures.
